# Effect of Microplastics on the Activity of Digestive and Oxidative-Stress-Related Enzymes in Peled Whitefish (*Coregonus peled* Gmelin) Larvae

**DOI:** 10.3390/ijms241310998

**Published:** 2023-07-01

**Authors:** Yulia A. Frank, Elena A. Interesova, Mikhail M. Solovyev, Jiayi Xu, Danil S. Vorobiev

**Affiliations:** 1Biological Institute, Tomsk State University, Tomsk 634050, Russia; interesovaea@yandex.ru (E.A.I.);; 2Institute of Systematics and Ecology of Animals SB RAS, Novosibirsk 630091, Russia; 3State Key Laboratory of Estuarine and Coastal Research, East China Normal University, Shanghai 200241, China; jyxu@sklec.ecnu.edu.cn

**Keywords:** microplastics, PS microspheres, freshwater fish, peled whitefish, fish larvae, enzymatic activity, digestive enzymes, antioxidant enzymes

## Abstract

Microplastics (MPs) are emergent pollutants in freshwater environments and may impact aquatic organisms, including those of nutritional value. The specific activities of digestive and antioxidant enzymes can be used as good bioindicators of the potential effects of MPs on fish in case of waterborne MP contamination. In this study, we used fluorescent polystyrene microplastics (PS-MPs) to analyze the alterations in enzyme activities in larvae of *Coregonus peled* Gmelin (peled or Northern whitefish), one of the most valuable commercial fish species of Siberia. Our results indicate that peled larvae can ingest 2 µm PS microspheres in a waterborne exposure model. A positive correlation (*r_s_* = 0.956; *p* < 0.01) was found between MP concentration in water and the number of PS microspheres in fish guts, with no significant differences between 24 h and 6-day exposure groups. The ingestion of MPs caused alterations in digestive enzyme activity and antioxidant responses at the whole-body level. The presence of PS-MPs significantly stimulated (*p* < 0.05) the specific activity of α-Amylase and non-specific esterases in peled larvae after 24 h. However, a pronounced positive effect (*p* < 0.05) of MPs on the activity of pancreatic trypsine and bile salt-activated lipase was only found after 6 days of exposure compared to after 24 h. Intestinal membrane enzyme aminopeptidase N was also stimulated in the presence of PS-MPs after 6-day exposure. We also observed a significant increase in the specific activity of catalase in peled larvae after 6 days of exposure, which indicates the MP-induced modulation of oxidative stress. Taken together, these results highlight the potential impact of environmental MPs on northern commercial fish, their importance for estimating fish stocks, and the sustainability of freshwater ecosystems.

## 1. Introduction

Environmental pollution caused by microplastics (MPs), small plastic particles of various chemical compositions, shapes, colors, and polymer matrices, is a growing concern [1]. The problem is acute due to the global increase in plastic production and due to the limited biological and chemical degradability of microplastics, which leads to their rapid accumulation in ecosystems [2]. The ability of organisms across trophic levels to uptake MPs was documented [3]. Recently, there has been emerging evidence of MPs in commercial wild fish and fishmeal [4], highlighting the issue of MP contamination and its potential implications for human health [5,6]. However, our understanding of the short- and long-term effects of MPs on aquatic organisms, particularly wild fish and ecosystems, is still limited.

Freshwater fish, considered as a key receptor and bioindicator of MP pollution in rivers and lakes, provide valuable models to expand knowledge on how MPs affect the ecology and physiology of animals [7]. Published data indicate that MPs can have broad implications for freshwater fishes at almost all biological levels, including cellular, organ/tissue, individual, population, community, and ecosystem levels, inducing metabolic alterations, mortality, and changes in population and ecosystem structures [8]. 

Various biomarkers are used to assess the adverse effects of toxic substances on animals. Digestive enzymes play a crucial role in assessing the impact of toxicants on the gastrointestinal tract (GIT), as alterations in their activity can inevitably cause metabolic disorders. MPs can enter the bodies of aquatic animals, including fish, through several pathways: (1) predators mistaking particles for prey; (2) organisms passively absorbing MPs during nutrient filtration; (3) organisms accidentally absorbing MPs from the environment through non-filtering diets; (4) organisms receiving MPs through food chains, and (5) organisms passively absorbing MPs from the environment during respiration [9,10,11,12,13,14,15]. Thus, four out of the five probable pathways of MP uptake in aquatic animals suggest the initial entry of particles into the digestive tract, which indicates that the potential impacts of MPs on the physiological processes may primarily be associated with digestion. It is now known that MPs can adversely affect the GIT of aquatic animals, causing damage to the inner surface of the organs [16,17,18,19,20,21,22,23,24,25,26], altering intestine microbiota [21,27,28,29,30,31,32], and disrupting digestion [33,34]. Recent studies have indicated that MPs can also change the activity of digestive enzymes in freshwater fish [25,35,36].

To assess the general state of an organism, the activity of oxidative-stress-related antioxidant enzymes can be used as an indicative biomarker to reveal physiological responses to reactive oxygen species (ROS) generation that causes oxidative stress-induced pathology and cell and tissue damage [37,38]. ROS production and antioxidant enzyme activity alterations are MP-related physiological responses commonly observed in fish [39]. GIT oxidative stress along with histological damage are the most reported organ and level effects of MPs in freshwater fish [7]. The implications of MPs regarding the activity of enzymes involved in protecting cells from oxidative damage, such as catalase, superoxide dismutase, and glutathione peroxidase, were evaluated in *Danio rerio* [21,24,28,40] and several commercial freshwater species [41,42], yielding controversial results. 

The aim of this study was to analyze the effect of MPs on digestive and antioxidant enzymes in larvae of *Coregonus peled* Gmelin (peled or Northern whitefish), one of the most commercially valuable and commonly consumed fish species in the Ob River basin. Peled is an exquisite dish and a traditional source of protein nutrition for the indigenous people of the Russian North. However, the abundance of this species has significantly decreased due to overfishing when allowable catch quotas are exceeded and deteriorating water quality [43]. Due to plastic pollution in the Ob River system [44,45,46], MPs can become an additional factor contributing to the negative impact on peled populations, with larvae and juveniles of this valuable fish species being particularly at risk.

## 2. Results and Discussion

Changes in the specific activities of the range of digestive and antioxidant enzymes associated with PS microspheres were evaluated in 3-week-old peled larvae (Figure 1A). The initial activity of the enzymes was analyzed before the experiment, and then under control conditions without MPs, as well as in the presence of PS microspheres at concentrations of 5, 50, and 500 µg L^−1^ after 24 h (1 day) and 6 days of exposure to assess potential alterations. PS was chosen as a model MP since it is one of the most widely produced plastics globally, accounting for more than 5% of global plastic production [47]. Moreover, micro- and nanosized particles of PS have been used in numerous experiments to assess plastic implications in freshwater fish [21,23,25,26,28,35,41], allowing for meaningful comparisons of results. The chosen exposure concentrations of 5–500 μg L^−1^ are environmentally relevant and consistent with previously published data on the toxicity of MPs to freshwater fish, which typically ranges from 1 to 1000 μg L^−1^ [20,21,25,27,48].

The presence of MPs in peled larvae from the experimental groups was confirmed microscopically and quantified. After exposure to PS-MPs, the yellow-green, fluorescent 2 μm spheres were clearly visible inside the examined individuals. No particles were observed in larvae from the control groups both after 24 h and after 6 days. Fluorescent particles were detected in the groups exposed to 50 and 500 μg L^−1^ for 24 h, accounting for 8.30 ± 3.12 and 107 ± 18.8 items per individual^−1^, respectively (Figure 1B–D). Only one MP was found in 10 randomly selected individuals from the 5 μg L^−1^ group, which is equivalent to 0.10 ± 0.31 items per individual^−1^. The average number of PS-MPs detected in the fish gut after 6 days varied from 0.2 ± 0.42 items per individual^−1^ at the lowest MP concentration in the 5 μg L^−1^ group to 10.9 ± 4.33 and 116 ± 34.4 items per individual in the 50 and 500 μg L^−1^ groups, respectively, with no significant differences compared to that detected after 24 h of exposure (*p* > 0.05). The localization of PS spheres in the GITs of peled larvae exposed to 5, 50, and 500 μg L^−1^ of PS-MPs indicates that plastic particles were ingested by fish larvae and retained in the gut for some time. A strong positive correlation (*r_s_* = 0.956, *p* < 0.01) was found between the nominal MP concentration level and the number of PS microspheres in fish guts. 

### 2.1. Effect of PS Microspheres on Digestive Enzymes

Given that MPs primarily come into contact with the GIT upon ingestion, they are expected to induce a physiological response at the level of digestive enzymes. Although less data are available on this group of enzymes compared to enzymes that reflect the overall state of the body, several studies have confirmed alterations in their activity in response to the direct or indirect impacts of MPs on freshwater fish [25,36,48,49,50].

In this study, we evaluated alterations in the activity (in terms of U g^−1^ of protein) of pancreatic enzymes, including trypsin (TRP, E.C. 3.4.21.4), chymotrypsin (CHY, EC. 3.4.21.1), carboxypeptidase A (CPA, EC. 3.4.17.1), α-Amylase (AMY, E.C. 3.2.1.1), and bile salt-activated lipase (BAL, E.C. 3.1.1.-), as well as intestinal (brush border) alkaline phosphatase (ALP, E.C. 3.1.3.1) and aminopeptidase N (APN, EC 3.4.11.2), and non-specific esterases (NSE, E.C. 3.1.-), which can have mixed origins (pancreatic and intestinal) in the presence of PS microspheres. To assess these alterations, the initial activity of digestive enzymes was analyzed before the experiment (referred to as ‘0 days’ in Figure 2). Subsequently, the activity of digestive enzymes in peled larvae was determined under control conditions without MPs and in the presence of PS microspheres after 24 h and after 6 days of exposure. Nominal PS-MP exposure concentrations were 5, 50, and 500 µg L^−1^ (Figure 2). 

The specific activity of TRP showed a clear tendency to increase from the control group to the group exposed to 500 µg L^−1^ PS-MPs, as shown in Figure 2. Yet, the specific activities of other digestive enzymes—pancreatic CHY, AMY and BAL, as well as intestinal ALP and APN—showed inconsistent changes at first glance. For four out of eight assayed enzymes, a significant correlation was found between the specific activity and experimental parameters (Figure 2). A moderate correlation was observed between CPA activity and MP concentration (*r*_s_ = 0.596, *p* < 0.01). A weak to strong correlation between enzymatic activity and exposure time was observed for BAL (*r*_s_ = 0.385, *p* < 0.01) and for APN (*r*_s_ = 0.675, *p* < 0.01). Only TRP activity was significantly dependent on both MP concentration and exposure time (*r*_s_ = 0.406, *p* < 0.01 and *r*_s_ = 0.328, *p* < 0.05, respectively).

The comparison of the specific activity of digestive enzymes between the experimental and control groups, as well as between the experimental groups after 6 days and after 24 h of exposure, revealed several significant findings. AMY exhibited a significant increase in activity at every tested PS-MP concentration after 24 h (*p* < 0.05), and a similar trend was found for pancreatic TRP after 6 days (*p* < 0.01) (Table 1). The specific activity of NSE in peled larvae significantly increased (*p* < 0.05) compared to the control in the presence of 50 and 500 µg L^−1^ of MPs after 24 h of exposure. The activities of TRP and NSE, but not of AMY, significantly differed (*p* < 0.05) after 6 days compared to those after 24 h (Table 2). However, NSE activity was higher after 6-day exposure in the control, which indicated the physiological nature of the changes. The specific activity of BAL showed inconsistent changes at different PS microsphere concentrations after 24 h. However, a significant increase in BAL activity compared to that under control conditions was observed after 6 days of exposure at concentrations above 50 µg L^−1^ (*p* < 0.01). The activity of BAL was also significantly higher (*p* < 0.05) in the presence of MPs after 6 days of exposure compared to that detected after 24 h (Table 2).

A pronounced positive overall effect of MPs was noted for pancreatic CPA and intestinal ALP and APN after 6 days of exposure compared to the control (Table 1). However, differences in APN activity in the presence of PS-MPs were only significant (p < 0.01) after 6 days compared to those observed after 24 h. Generally, 6-day exposure of peled larvae to PS-MPs caused more pronounced alterations in the activity of digestive enzymes compared to the acute 24 h experiment. The specific activity of five out of eight studied pancreatic and intestinal enzymes in the presence of MPs, including TRP, CPA, BAL, ALP, and APN, significantly differed from the control after 6 days (Table 1). TRP, BAL, and APN activities were significantly higher in the presence of PS-MPs after 6 days of exposure compared to those found after 24 h (Table 2).

Our results indicate that the specific activity of pancreatic enzymes (TRP, CPA, AMY, BAL) and NSE in peled larvae was significantly stimulated in the presence of PS-MPs after 1 or 6 days of exposure. In contrast, the study by Huang et al. [50] reported a decrease in TRP, CHT, AMY and non-specific lipase activity in juvenile guppies exposed to 32–40 μm PS-MPs at concentrations of 100–1000 µg L^−1^ in a long-term (28 days) experiment. Wen et al. [48] also found a decrease in the activity of TRP in juvenile *Symphysodon aequifasciatus* (discus fish) after 30 days of exposure to PS-MPs of the same size. Additionally, feeding of *Nothobranchius guentheri* with 5 and 15 μm PS-MPs reduced the activities of TRP, CHT, AMY and lipase in an extended experiment that lasted 42–46 weeks (*p* < 0.05) [36]. 

Similar to our results, the stimulation of digestive enzymes in the presence of 200 to 1000 µg L^−1^ polyvinyl chloride (PVC) was found in freshwater *Barbodes gonionotus* (silver barb) fry after 96 h of exposure [49]. Pancreatic TRP and CHT activities significantly increased in juvenile fish exposed to 500 and 1000 µg L^−1^ PVC fragments of 0.10 to 1.00 mm as compared to the control group without MPs and groups exposed to 200 µg L^−1^ PVC. TRP activities (the experimental/control ratio) detected in peled larvae in our study were several-fold lower compared to those found for silver barb after 4 days of exposure. This difference could be attributed to shape-dependent accumulation effects, as previous studies have shown that MPs accumulate differently in zebrafish guts and induce intestinal injury based on their shape (fibers > fragments > spheres) [22]. Another possible explanation is a different age of the fish, as digestive enzyme activities vary significantly during the early stages of ontogeny and can be affected by factors such as species, diet, and feeding regime [51]. 

The uptake and physical impact of MPs is well established to induce various functional and anatomical alterations in GITs, which can cause malnutrition and developmental problems [17,35,52,53]. Pancreatic hydrolytic enzymes are able to break down biopolymers and enforce nutrient absorption. The decrease in pancreatic hydrolytic enzymes TRP, CHT, AMY and lipase in the gut of juvenile guppies was attributed to the presence of MPs in fish guts and reduced sensitivity to food [50]. The increased specific activity of pancreatic enzymes revealed in this study could also be a consequence of impaired nutrient absorption caused by the impact of MP on fish GIT. The thickening of the intestinal epithelium [49] or disruption of intestinal folds and reduction in the intestinal surface area due to the physical presence of MPs [20] can decrease nutrient absorption. A significant increase in TRP and CHT production under limited food conditions was previously detected in sea bass larvae [54]. Another study reported that the activity of TRP in common carp subjected to a short 2-day starvation was higher than that in fish fed normally [55].

The contradictions in specific activity alterations of pancreatic digestive enzymes described above can be attributed to exposure duration. Short-term MP exposure to GIT, as in our experiment (24 h to 6 days) or a 96 h exposure in the study by Romano et al. [49], may stimulate enzymatic activity, while a chronic long-term exposure lasting several weeks [48,50] may lead to further suppression of enzymatic activity. An increase in the specific activity of intestinal ALP and exopeptidase APN, induced by short-term exposure in the current study, may be due to the direct impact of MPs on the gut epithelium. It was previously shown that intestinal crypt and villi cell loss significantly increased in *Girella laevifrons* due to physical abrasion after the exposure of juveniles to Poly(styrene-co-divinylbenzene) MPs [56]. 

### 2.2. Effect of PS Microspheres on Antioxidant Enzymes

Microplastics are known to induce toxicity through the formation of free radicals, which cause damage to cellular macromolecules and subsequently lead to physiological and biochemical changes in animals [25,57]. However, currently available quantitative data on the effect of MPs on antioxidant enzyme activity in fish are largely contradictory. Some studies showed an increase in the activity of superoxide dismutase (SOD) in fish exposed to MPs [21,28,41,42], while others reported a decrease in SOD activity in freshwater fish [36]. In our study, we analyzed important biomarkers related to oxidative stress defense in cells, including catalase (CAT, E.C. 1.11.1.6), glutathione S-transferase (GST, E.C. 2.5.1.18), glutathione reductase (GR, E.C. 1.8.1.7), and glutathione peroxidase (GPx, E.C. 1.11.1.9). 

Changes in the activity (U mg protein^−1^) of antioxidant enzymes in peled larvae exposed to PS microspheres were evaluated. Similar to the digestive enzyme experiment, the initial activity of antioxidant enzymes was analyzed before the experiment (‘0 days’), and then, the activity of the enzymes was assessed under control conditions without MPs and in the presence of PS microspheres (at concentrations of 5, 50, and 500 µg L^−1^) after 24 h and after 6 days of exposure to assess possible alterations. 

In the current study, the specific activity of the most assayed antioxidant enzymes in peled larvae showed an increasing trend in the presence of PS-MPs after 1 and/or 6 days of exposure (Figure 3). A contradictory effect of MPs was revealed for GPx. In the case of short-term exposure, enzymatic activity decreased, while after 6 days, it was activated compared to that under control conditions (Figure 3). A moderate correlation between enzymatic activity and MP concentration, as well as between enzymatic activity and exposure time, was observed for CAT (*r*_s_ > 0.4, *p* < 0.01), and a weak to moderate correlation (*r*_s_ = 0.309, *p* < 0.05 and *r*_s_ = 0.514, *p* < 0.01) was observed for GST (Figure 3). The activity of GPx weakly correlated (*r*_s_ = 0.305, *p* < 0.05) with the MP level; GR activity was strongly dependent on exposure time (*r*_s_ > 0.645, *p* < 0.01). 

We observed a consistent significant response in CAT activity in peled larvae after 6 days of MP exposure (Table 3 and Table 4). CAT along with SOD acts as the primary defense mechanism against MP-induced oxidative stress through the conversion of superoxide radicals to hydrogen peroxide and finally to oxygen and water. It is believed that the production of reactive oxygen species triggered by MP exposure in fish stimulates antioxidant responses and increases antioxidant enzyme activity [25]. However, conflicting results regarding alterations in antioxidant enzyme activity in fish under MP exposure have been reported in the literature. In several experiments, the specific activity of CAT increased under PS-MP exposure in zebrafish, guppy, and discus fish [21,28,48,50]. Some other studies reported a decrease in CAT activity in various freshwater fish species in response to MPs [36,40,42,58]. Analysis of the literature did not show clear associations between CAT activity and the design of the experiment or the type/sizes of MPs used to assess the effect of waterborne MP exposure on fish. A series of experiments performed using standardized protocols would help resolve these contradictions.

Significant differences in enzymatic activity in species from the experimental group compared to the control group were detected for GST and GR both after 24 h and after 6 days of exposure, and for GPx after 6 days of exposure (Table 3). GST is involved in the second phase of the detoxification process catalyzing the binding of glutathione, which plays an important role in maintaining the redox state of exogenous substances to prevent damage to biomolecules. GPx assists hydrogen peroxide conversion to less toxic compounds and leads to glutathione oxidation, which is subsequently reduced by GR [25].

In our study, GST activity in peled larvae showed a significant increase (*p* < 0.05) after acute (24 h) exposure to PS-MPs at concentrations higher than 50 µg L^−1^ compared to the control group (Table 3). Similarly, GR activity significantly increased (*p* < 0.05) in the experimental groups exposed to MP at concentrations of 5, 50, and 500 µg L^−1^ and in the presence of PS-MP as a whole (*p* < 0.01) after 24 h of exposure. On the contrary, the significant stimulation (*p* < 0.01) of GST and GR activities after 6 days compared to that after 24 h was detected under both experimental and control conditions (Table 4). 

The specific activity of GPx significantly decreased in the 50 µg L^−1^ group and in the presence of PS-MPs as a whole compared to that in the control (Table 3). A significant decrease (*p* < 0.05) in GPx activity after 6 days compared to that after 24 h was observed under both experimental and control conditions (Table 4).

In most studies of the MP waterborne impact on freshwater fish, the specific activity of GST and GR in organs and tissues significantly increased [42,50,58]. However, contradictory results were reported for the specific activity of GPx in freshwater fish juveniles and adults. GPx activity either significantly increased in response to MP exposure, facilitating peroxide conversion to non-toxic hydroxyl [42,48], or decreased [36,59,60]. Umamaheswari et al. [55] also reported a decrease in GPx activity in *D. rerio* under PS-MPs exposure.

## 3. Materials and Methods

### 3.1. Object of the Study 

*C. peled* (Gmelin, 1789), peled or northern whitefish, inhabits rivers and lakes in Northern Eurasia, ranging from the Mezen River to the Kolyma River [61]. The peled population is most abundant in the Ob River basin, where it undertakes long migrations. During the winter, it resides in the Gulf of the Ob, and then moves to the floodplain system of the lower reaches of the river to fatten up. By autumn, it migrates to the upper part of the basin for reproduction, seeking out suitable sandy-pebble bottom areas for spawning [62]. Peled is an important fish in Siberia [63,64,65]. 

Three-week-old larvae of *C. peled* with similar body weights (wet weight: 4.16 ± 0.70 mg, mean ± SD) were used for the experiments. Peled larvae were bred under laboratory conditions from the egg mass of wild fish in the Research and Production Company ‘Tomsk-Ecologija’ SJC (Tomsk, Russia), which specializes in the Siberian River ichthyofauna restoration.

### 3.2. MP Preparation

Fluorescent yellow-green carboxylate-modified PS microspheres with a size of 2.0 ± 0.2 µm (Sigma-Aldrich, St. Louis, MO, USA) with the excitation/emission wavelengths of 470/505 nm were used in the experiments. MPs were stored in the dark as an aqueous suspension with a density of 1.050 g cm^−3^. The working solution of the microspheres was prepared as described in [66]. Briefly, the stock solution of MPs was thoroughly mixed using a vortex (Microspin BioSan FV-2400, Riga, Latvia) to spread particles more evenly. MPs were then subjected to three cycles of washing. For each cycle, 1 mL of the stock solution was centrifuged at 9000 rpm for 10 min, the supernatant was carefully discarded, and 1 mL of distilled water filtered with a 0.22 µm modified cellulose membrane NEWSTAR (Hangzhou Special Paper Industry Co., Hangzhou, China) was added. After each washing step, the PS microspheres were resuspended via vortexing.

### 3.3. Experimental Design

The experiments were conducted in an aerated 25 L glass aquarium filled with 20 L of tap water. The water was circulated via a filtered water pump, and the experiments were conducted under diffuse light, at a temperature of 13.0 ± 0.5 °C, pH of 7.5 ± 0.1, and dissolved oxygen concentration of 11.5 ± 0.5 mg L^−1^. Larvae were fed with Coppens Advance 0.2–0.3 mm dried fish food every 2 h during the entire period.

Treatment was performed in 4 modes: (1) a control group with no MPs, (2) treatment with 5 µg L^−1^ PS-MPs added, (3) treatment with 50 µg L^−1^ PS-MPs added, (4) treatment with 500 µg L^−1^ MPs added. Each treatment group contained about 2000 *C. peled* larvae (~100 individuals L^−1^). The control and treatment groups were randomly assigned to positions on the laboratory bench to reduce the experimental error. Three replicates in each group were sampled in a random order after 24 h and after 6 days of exposure (the maximum possible cultivation period for peled larvae in the no-flow system). After sampling, each replicate of 50 specimens allotted for enzyme activity assays was weighed. After being sacrificed, the sampled individuals were rinsed with distilled water and conserved at −80 °C until further analyses. The rest of the specimens from each replicate were joined into one sample for each control or experimental group, weighted and stored frozen until microscopy. 

### 3.4. Microscopy and Imaging

The ingestion and distribution of PS-MPs were observed using direct epifluorescence microscopy on 10 washed intact peled larvae randomly selected from each experimental group and counted. To determine the presence of the MPs and capture the images, a fluorescent microscope Axio Zoom.V16 (Carl Zeiss, Oberkochen, Germany) equipped with 38 HP eGFP filter set (Carl Zeiss) with excitation/emission wavelengths of 450–490/500–550 nm was used. Light microscopy was applied to observe the peled larvae in the experiments using the same microscope. The ZEISS Axiocam 712 mono digital camera and ZEN (Blue edition) 1.0 software was used to capture microphotographs of the fish larvae.

### 3.5. Enzyme Activity Assays 

#### 3.5.1. Sample Preparation

All larvae were completely homogenized for analytical purposes, since they were too small to be dissected [67]. Digestive enzyme activities were determined using 50–100 specimens per biological replicate at each experimental point. For quantifying the activity of oxidative stress and pancreatic enzymes (trypsin, chymotrypsin, carboxypeptidase A, α-Amylase, and bile salt-activated lipase) and non-specific esterases, samples were prepared according to the recommendations by Gisbert et al. [68]. The samples were homogenized (Ultra-Turrax T10 basic, IKA©-Werke, Germany) in 30 volumes (*v*/*w*) of cold 50 mM mannitol, 2 mM Tris-HCl buffer (pH 7.0) and centrifuged at 9000× *g* for 10 min at 4 °C; the required portion of supernatant was removed for enzyme quantification and kept at −80 °C until further analysis. The remaining supernatant was used to evaluate the activity of intestinal brush border membrane enzymes (alkaline phosphatase and N-aminopeptidase) after two-step purification [68]. In order to minimize the influence of freeze/thaw cycles, the supernatant was allotted and stored at −80 °C until quantification [69].

#### 3.5.2. Digestive Enzymes

The activity of trypsin (E.C. 3.4.21.4) was assayed using BAPNA (N-α-benzoyl-DL-arginine p-nitroanilide) as substrate. One unit of trypsin per ml (U) was defined as 1 μmol BAPNA hydrolyzed per min^−1^ mL^−1^ of enzyme extract at 407 nm [70]. Chymotrypsin (EC. 3.4.21.1) activity was quantified using BTEE (benzoyl tyrosine ethyl ester) as a substrate, and its activity (U) corresponded to 1 μmol BTEE hydrolyzed per min^−1^ mL^−1^ of enzyme extract at 256 nm [71]. Carboxypeptidase A (EC. 3.4.17.1) activity was estimated using hippuryl-L-phenylalanine as substrate, and its activity (U) corresponded to 1 μmol hippuryl-L-phenylalanine hydrolyzed per min^−1^ mL^−1^ of enzyme extract at 254 nm [72]. α-Amylase (E.C. 3.2.1.1) activity was determined according to Bernfeld [73] using 0.3% soluble starch. The enzyme extract was incubated with starch for 120 min. After that, the reaction was stopped using 3.5-dinitrosalicylic acid and incubated at 100 °C for 10 min. Different concentrations of maltose were used as a standard curve. The activity of amylase activity (U) was defined as 1 mg of maltose liberated per min^−1^ mL^−1^ of enzyme extract at 540 nm. Bile salt-activated lipase (E.C. 3.1.1.-) activity was assayed using p-nitrophenyl myristate as a substrate. One unit of lipase per ml (U) was defined as 1 μmol p-nitrophenyl myristate hydrolyzed per min^−1^ mL^−1^ of enzyme extract at 405 nm [74]. Non-specific esterases (E.C. 3.1.-) activity was assayed using p-nitrophenyl acetate as substrate. One unit of esterases per ml (U) was defined as 1 μmol p-nitrophenyl acetate hydrolyzed per min^−1^ mL^−1^ of enzyme extract at 405 nm [75]. 

Regarding intestinal enzymes, alkaline phosphatase (E.C. 3.1.3.1) was quantified using 4-nitrophenyl phosphate (PNPP) as substrate. One unit (U) was defined as 1 μmol PNPP hydrolyzed per min^−1^ mL^−1^ of enzyme extract at 407 nm [76]. Aminopeptidase N (EC 3.4.11.2) activity was determined according to Maroux et al. [77] using sodium phosphate buffer 80 mM (pH 7.0) and L-leucine p-nitroanilide as a substrate in DMSO. One unit of enzyme activity (U) was defined as 1 μg nitroanilide released per min^−1^ mL^−1^ of enzyme extract and measured at 410 nm.

#### 3.5.3. Antioxidant Enzymes

Homogenized samples (after the first step of homogenization) were used to measure antioxidant enzyme activities. Catalase (E.C. 1.11.1.6) activity was estimated in larvae samples via the decrease in absorbance at 240 nm, using H_2_O_2_ as a substrate [78]. Glutathione S-transferase (E.C. 2.5.1.18) activity was assayed by the formation of glutathione chlorodinitrobenzene adduct at 340 nm, using 1-chloro-2,4-dinitrobenzene and glutathione as substrates [79]. Glutathione reductase (E.C. 1.8.1.7) activity was determined by measuring the oxidation of nicotinamide adenine dinucleotide phosphate reduced (NADPH) at 340 nm, using glutathione disulphide and NADPH as substrates [80]. Total glutathione peroxidase (E.C. 1.11.1.9) was determined according to Günzler and Flohé [81] by measuring the consumption of NADPH at 340 nm, using glutathione and NADPH as substrates.

All the measurements were made at an ambient temperature in three biological replicates with three methodological replicates from each pool of larvae and absorbance read using a spectrophotometer PowerWave XS2 (BioTek, Minneapolis, MN, USA). Enzymatic activities were expressed as specific activity defined as units per gram of protein (U g protein^−1^) for digestive enzymes and as units per milligram of protein (U mg protein^−1^) for antioxidant enzymes [82] using bovine serum albumin as standard.

### 3.6. Data Interpretation and Statistical Analysis

When interpreting the experimental data, we estimated the activity of digestive and antioxidant enzymes in the peled larvae under control conditions (absence of MPs) and in the presence of PS-MPs at concentrations of 5, 50, and 500 µg L^−1^. Data were collected at baseline (0 days) and after 1 and 6 days of exposure. The data on enzyme activity assays per U g protein^−1^ or U mg protein^−1^ were quantified as the arithmetic mean ± standard error (SE). 

Differences in the activity of digestive enzymes between peled larvae in the presence of PS-MPs (5, 50, and 500 µg L^−1^) and those from the control, as well as the overall differences across all MP concentrations, were evaluated. A nonparametric Mann–Whitney U-test [83] was used to determine significant differences (*p* < 0.01 and <0.05) between the control group and MP-treated specimens. The correlation between the experimental parameters and MP counts in fish guts or between the experimental parameters and enzymatic activities was evaluated using the Spearman rank correlation coefficient (*r_s_*) [84].

## 4. Conclusions

Our results show that 2 µm PS microspheres can be ingested by peled (*C. peled*) larvae in the model of waterborne exposure and cause alterations in the activity of digestive enzymes and in antioxidant responses at the whole-body level. 

The number of ingested MPs was strongly dependent on the particle concentration (*r_s_* = 0.956, *p* < 0.01). At the same time, no significant differences were detected in the number of PS microspheres in fish guts between 24 h and 6-day exposure groups. 

The specific activity of pancreatic digestive enzyme (α-Amylase) and non-specific esterases in peled larvae was significantly stimulated (*p* < 0.05) in the presence of PS-MPs after 24 h. However, a pronounced stimulating effect of MPs on the activity of pancreatic trypsine, carboxypeptidase A, and bile salt-activated lipase compared to the control groups was observed only after 6 days of exposure (*p* < 0.01). Intestinal membrane enzymes, alkaline phosphatase and aminopeptidase N, were also stimulated in the presence of PS-MPs after 6-day exposure. However, only the activities of trypsine, lipase and aminopeptidase N were significantly higher (*p* < 0.05) in the presence of PS-MPs after 6 days compared to those after 24 h of exposure.

A comparison of the data obtained with the results of other studies on freshwater fish, including those assessing the chronic effects of MPs, suggests that short-term MP exposure (from 24 h to 5–7 days) may stimulate the activity of pancreatic hydrolytic enzymes. In contrast, a chronic effect found in long-term (>14 days and several weeks) experiments leads to the further suppression of enzymatic activity. It is evident that prolonged MP consumption poses greater risks to individual fish species in terms of the general physiological and biochemical effects of MPs on the GIT and freshwater food webs.

Exposure of freshwater fish to MPs can either stimulate or inhibit antioxidant reactions and induces ROS production. MPs affect glutathione and its response cycles in fish, contributing to antioxidant reactions. In the current study, specific activity of the most assayed antioxidant enzymes in peled larvae showed an increasing trend in the presence of PS-MPs. Significant but contradictory differences in enzymatic activity compared to the control group were detected for GST, GR and GPx after acute (24 h) PS-MP exposure. Generally, the specific activity of the four antioxidant enzymes, CAT, GST, GPx and GR, significantly differed (*p* < 0.05) from those in the control group after a longer (6-day) exposure in the presence of PS-MPs. Yet, a significant increase in catalase specific activity was only observed after 6-day exposure, while changes in the activities of glutathione S-transferase, glutathione reductase, and glutathione peroxidase were physiological. The data on the alteration of antioxidant enzyme activity under MP exposure obtained in our study and those from previous investigations are contradictory. Additional experiments performed using standardized protocols and an increased number of biological and analytical replicates can help resolve these contradictions.

The current study conducted on larvae of the previously untested commercial fish species *C. peled* contributes to the scientific understanding of the effects of MPs on physiology and biochemistry of freshwater fish. The results of this study are crucial for evaluating the effects of MPs on ecological sustainability and food security, particularly in northern regions that are highly dependent on water resources.

## Figures and Tables

**Figure 1 ijms-24-10998-f001:**
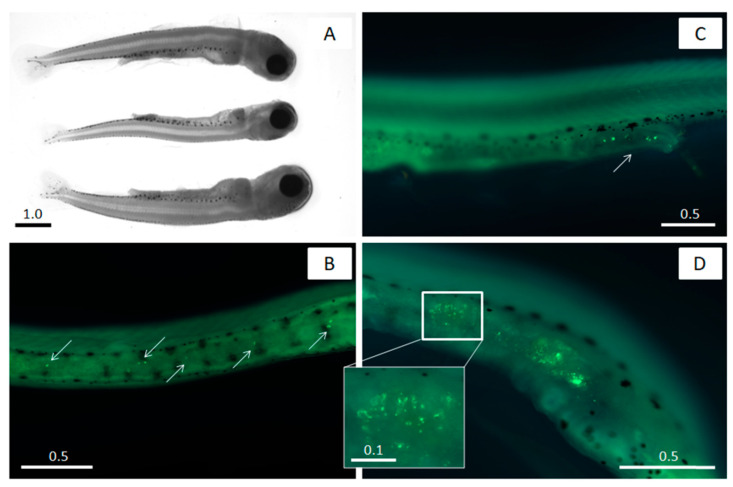
Microphotograph of the peled larvae used for the experiments (**A**), and the presence of fluorescent PS-MPs in the gut of peled larvae after exposure to 50 μg L^−1^ (**B**) and 500 μg L^−1^ (**C**,**D**) MPs. The arrows show the localization of microplastic particles. Scale bars are in mm.

**Figure 2 ijms-24-10998-f002:**
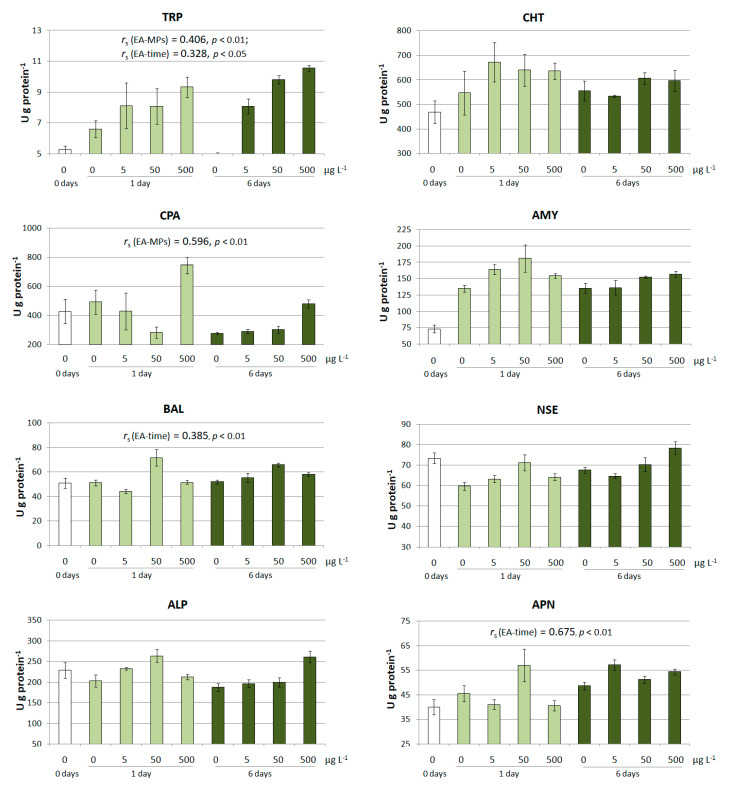
Initial activity of digestive enzymes in peled larvae (0 days). Activity of the enzymes in the control (0 µg L^−1^) and in the presence of PS-MPs at concentrations of 5, 50, and 500 µg L^−1^ after 24 h (1 day) and after 6 days of exposure. Data are expressed as the mean ± standard error (*n* = 3). Significant Spearman correlation coefficients (*r*_s_) between enzyme activities (EA) and MP concentration (MPs) or between EA and exposure time are shown. Enzyme designations (hereinafter): TRP, trypsin; CHT, chymotrypsin; CPA, carboxypeptidase A; AMY, α-Amylase; BAL, bile salt-activated lipase; NSE, non-specific esterases; ALP, alkaline phosphatase; APN, aminopeptidase N.

**Figure 3 ijms-24-10998-f003:**
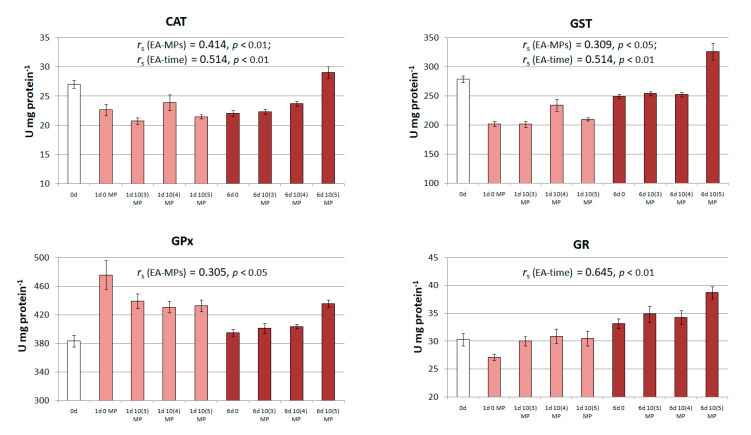
Initial activity of antioxidant enzymes in peled larvae (0 days), activity of the enzymes in the control (0 µg L^−1^) and in the presence of PS-MPs at concentrations of 5, 50, and 500 µg L^−1^ after 24 h (1 day) and after 6 days of exposure. Data are expressed as the mean ± standard error (*n* = 3). Significant Spearman correlation coefficients (*r*_s_) between enzymatic activity (EA) and MP concentration (MPs) or between EA and exposure time are shown. Enzyme designations (hereinafter): CAT, catalase; GST, glutathione S-transferase; GPx, glutathione peroxidase; GR, glutathione reductase.

**Table 1 ijms-24-10998-t001:** Differences in the activity of digestive enzymes in peled larvae in the presence of PS-MPs at concentrations of 5, 50, and 500 µg L^−1^ and in the presence of PS-MP as a whole (ALL) after 24 h (1 day) and 6 days of exposure compared to the control.

6 days								
500 µg L^−1^	0.01	no	0.01	0.05	0.01	0.01	0.01	0.01
50 µg L^−1^	0.01	no	0.01	0.05	0.01	no	no	no
5 µg L^−1^	0.01	no	no	no	no	no	no	0.01
ALL	0.01	NO	0.01	NO	0.01	NO	0.05	0.01
	**TRP**	**CHT**	**CPA**	**AMY**	**BAL**	**NSE**	**ALP**	**APN**
ALL	NO	NO	NO	0.01	NO	0.05	NO	NO
500 µg L^−1^	0.01	no	0.05	0.01	no	0.05	no	no
50 µg L^−1^	no	no	no	0.05	0.01	0.05	0.05	no
5 µg L^−1^	no	no	no	0.05	0.05	no	no	no
1 day								

Note: ‘no’—no differences, ‘0.01’—significant differences (*p* < 0.01), ‘0.05’—significant differences (*p* < 0.05). Red shading indicates inhibition of enzymatic activity compared to the control, and green shading indicates stimulation.

**Table 2 ijms-24-10998-t002:** Differences in the activity of digestive enzymes in peled larvae in the control, in the presence of PS-MPs at concentrations of 5, 50, and 500 µg L^−1^ and in the presence of PS-MP as a whole (ALL) after 6 days compared to those after 24 h (1 day) of exposure.

	TRP	CHT	CPA	AMY	BAL	NSE	ALP	APN
ALL	0.05	NO	NO	NO	0.05	0.05	0.05	0.01
500 µg L^−1^	no	no	0.01	no	0.01	0.01	0.05	0.01
50 µg L^−1^	no	no	0.05	no	no	no	0.01	no
5 µg L^−1^	no	no	no	0.05	0.05	no	0.01	0.01
CONTROL	0.05	NO	NO	NO	NO	0.01	NO	NO

Note: ‘no’—no differences, ‘0.01’—significant differences (*p* < 0.01), ‘0.05’—significant differences (*p* < 0.05). Red shading indicates inhibition of enzymatic activity compared to the control, and green shading indicates stimulation.

**Table 3 ijms-24-10998-t003:** Differences in the activity of antioxidant enzymes in peled larvae in the presence of PS-MPs at concentrations of 5, 50, and 500 µg L^−1^ and in the presence of PS-MP as a whole (ALL) after 24 h (1 day) and after 6 days of exposure compared to the control.

6 days				
500 µg L^−1^	0.01	0.01	0.01	0.01
50 µg L^−1^	0.01	no	no	no
5 µg L^−1^	no	no	no	no
ALL	0.01	0.05	0.05	0.05
	**CAT**	**GST**	**GPx**	**GR**
ALL	NO	0.05	0.05	0.01
500 µg L^−1^	no	0.05	no	0.05
50 µg L^−1^	no	0.05	0.05	0.05
5 µg L^−1^	no	no	no	0.01
1 day				

Note: ‘no’—no differences, ‘0.01’—significant differences (*p* < 0.01), ‘0.05’—significant differences (*p* < 0.05). Red shading indicates inhibition of enzymatic activity compared to the control, and green shading indicates stimulation.

**Table 4 ijms-24-10998-t004:** Differences in the activity of antioxidant enzymes in peled larvae in the control, in the presence of PS-MPs at concentrations of 5, 50, and 500 µg L^−1^ and in the presence of PS-MP as a whole (ALL) after 6 days compared to 24 h (1 day) of exposure.

	CAT	GST	GPx	GR
ALL	0.01	0.01	0.05	0.01
500 µg L^−1^	0.01	0.01	no	0.01
50 µg L^−1^	no	no	0.01	no
5 µg L^−1^	0.05	0.01	0.05	0.05
CONTROL	NO	0.01	0.01	0.05

Note: ‘no’—no differences, ‘0.01’—significant differences (*p* < 0.01), ‘0.05’—significant differences (*p* < 0.05). Red shading indicates inhibition of enzymatic activity compared to the control, and green shading indicates stimulation.

## Data Availability

Not applicable.

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
