# Peer review of "Effect of Microplastics on the Activity of Digestive and Oxidative-Stress-Related Enzymes in Peled Whitefish (Coregonus peled Gmelin) Larvae"

_ijms, 2023, doi:10.3390/ijms241310998_

Round 1

Reviewer 1 Report

General comment:

The present work analyzed the intestinal effect of microplastics in a cold-water fish species. However, as to the journal’s aim is focus on the molecular sciences, current data of enzyme activity is so limited. Additional experiments and demonstration of the underlying mechanism were required. Thus, the manuscript needs a major revision.

Specific comments:

1.  The villi cell loss is an important index for the effect of microplastics, as reported in previously study ("Microplastic ingestion cause intestinal lesions in the intertidal fish Girella laevifrons" published in Marine Pollution Bulletin). The villi structure and related function was not examined at either histological or molecular level in current study. Thus, the biological clues were not sufficient to support the mechanism underlined the altered activity of digestive and oxidative stress-related enzymes.

2.      Figure 1: There lacks the quantitative analysis of the fluorescent signals. ImageJ should be used to calculate the particles number or area.

3.      Figure 2 & 4: There lacks the analysis of significant differences. Besides the effect caused by different concentrations, the comparison between 1st and 6 st day post exposure also should be carried out.

4.      Figure 3 & 5: Current Figure 3 and Figure 5 were not typical figure, which should contain diagrams or pictures. So, just revise their format to be tables.

5.      So, it is suggested to separate the description of result and discussion section. The biological meaning and related deduction were not sufficiently dissected.

6.      All “p”, represented p-value, should be italic.

7.      For all mathematical symbols, there should be a space either before or after the symbol. Please check these points throughout the text.

  1. The manuscript should be revised by an English native speaker. Some gramma mistakes still could be found in the text.

Author Response

Dear Reviewer,

Thank you so much for the careful review of our manuscript and fair comments. We have revised the figures and made corrections in the text.

Your valuable comments helped us improve the manuscript.

  1. The villi cell loss is an important index for the effect of microplastics, as reported in previously study ("Microplastic ingestion cause intestinal lesions in the intertidal fish Girella laevifrons" published in Marine Pollution Bulletin). The villi structure and related function was not examined at either histological or molecular level in current study. Thus, the biological clues were not sufficient to support the mechanism underlined the altered activity of digestive and oxidative stress-related enzymes.

Thanks for the comment. Indeed, this is an excellent index and we will definitely use this histological approach in our future work with Siberian fish larvae. The recommended paper has been cited and has contributed to the discussion of the results in the section on the effect of MPs on digestive enzyme activity.

  1. Figure 1: There lacks the quantitative analysis of the fluorescent signals. ImageJ should be used to calculate the particles number or area.

The number of particles was visually counted under an epifluorescence microscope in the remaining frozen larvae (10 individuals for each group). Quantitative data were summarized and added to the Results and Discussion section.

  1. Figure 2 & 4: There lacks the analysis of significant differences. Besides the effect caused by different concentrations, the comparison between 1stand 6 st day post exposure also should be carried out.

Thank you for your helpful comment. The corresponding analysis is presented in Tables 1-4 not to overload the Figures with many details.

The comparison between 1st and 6 th days post exposure was carried out, and two tables (Table 2 and Table 4) were added to the Results and Discussion section.

  1. Figure 3 & 5: Current Figure 3 and Figure 5 were not typical figure, which should contain diagrams or pictures. So, just revise their format to be tables.

Revised.

  1. So, it is suggested to separate the description of result and discussion section. The biological meaning and related deduction were not sufficiently dissected.

If possible, we'd rather leave the results and discussion together, as the rules of the journal allow it.

  1. All “p”, represented p-value, should be italic.

Corrected.

  1. For all mathematical symbols, there should be a space either before or after the symbol. Please check these points throughout the text.

Corrected.

The manuscript should be revised by an English native speaker. Some gramma mistakes still could be found in the text.

The manuscript was revised by a professional translator.

Reviewer 2 Report

A simple study on the effects of MP on selected enzymes of fish larvae. There are several issue that need considerations:

1. The experimental design is generally sound but the sampling number varied between 50 and 100 samples. This might affect the results if not analysed properly

2. Several issues on the results:

a. To show the mere presence of MP in the larvae is not enough, especially when you have an experimental design that involves different levels and time points. The amounts of MP must be included

b. Most results on enzyme activities are in the form of summary. This should be improved with detailed presentation of enzyme activities according to the different level and time points.

c. Since the experiment involves several levels and time points, correlation between the levels with enzyme activities must be determined. Similarly, the correlation between enzyme activity and time points. This is relatively a simple study that requires extensive analysis especially statistical analysis such as the correlation analysis to strengthen the manuscript

3. There are several other minor comments that could be found in the attached file

Author Response

Dear Reviewer,

Thank you very much for the positive evaluation of our work and helpful comments.

We have made corrections in accordance with the comments:

  1. The experimental design is generally sound but the sampling number varied between 50 and 100 samples. This might affect the results if not analysed properly

For enzyme analysis, the same numbers of larvae (50) were taken; the rest of them were used for microscopy and working out the methodology. Appropriate clarifications were added to the text in Section 3.3:

«After sampling, each replicate of 50 specimens allotted for enzyme activity assays was weighed. After being sacrificed, the sampled individuals were rinsed with distilled water and conserved at −80 °C until further analyses. The rest of the specimens from each replicate were joined into one sample for each control or experimental group, weighted and stored frozen until microscopy.»

2. Several issues on the results:

  • To show the mere presence of MP in the larvae is not enough, especially when you have an experimental design that involves different levels and time points. The amounts of MP must be included.

The number of particles was visually counted under an epifluorescence microscope in the remaining frozen larvae (10 individuals for each group). Quantitative data were added to the Results section and the corresponding paragraph was revised as follows:

«The presence of MPs in peled larvae from the experimental groups was confirmed microscopically and quantified. After exposure to PS-MPs, the yellow-green fluorescent 2-μm spheres were clearly visible inside the examined individuals. No particles were observed in larvae from the control groups both after 24 hours and after 6 days. Fluorescent particles were detected in the groups exposed to 50 and 500 μg L-1 for 24 hours accounting for 8.30 ±3.12 and 107 ±18.8 items per individual-1, respectively (Fig. 1, B–D). Only one MP was found in 10 randomly selected individuals from the 5 μg L-1 group which is equivalent to 0.10 ±0.31 items per individual-1. The average number of PS-MPs detected in the fish gut after 6 days varied from 0.2 ±0.42 items per individual-1 at the lowest MP concentration in the 5 μg L-1 group to 10.9 ±4.33 and 116 ±34.4 items per individual-1 in the 50 and 500 μg L-1 groups, respectively, with no significant differences compared to that detected after 24 hours of exposure (p >0.05). The localization of PS spheres in the GITs of peled exposed to 5, 50 and 500 μg L-1 of PS-MPs indicates that plastic particles were ingested by fish larvae and retained in the gut for some time. Strong positive correlation (rs = 0.956, p <0.01) was found between the nominal MP concentration level and the number of PS microspheres in fish guts.»

  • Most results on enzyme activities are in the form of summary. This should be improved with detailed presentation of enzyme activities according to the different level and time points.

A detailed presentations of enzyme activities with respect to different level (5, 50 and 500 μg L-1) and time points (0, 1 and 6 days) are shown in Fig. 2 (for digestive enzymes) and in Fig. 3 (for antioxidant enzymes). Tables 1–4 summarize the differences between the experimental groups (different MP levels) and the control group at each time point.

  • Since the experiment involves several levels and time points, correlation between the levels with enzyme activities must be determined. Similarly, the correlation between enzyme activity and time points. This is relatively a simple study that requires extensive analysis especially statistical analysis such as the correlation analysis to strengthen the manuscript

Thank you for your helpful comment. Spearman rank correlation coefficients were calculated between MP concentrations and enzymatic activities, and between enzymatic activities and time points. Significant correlation coefficients of moderate and higher levels were displayed in Figs. 2 and 3 and discussed in the text.

As was suggested by the other Reviewer, we have also applied the Mann-Whitney U-test to compare 1st and 6 th days post exposure. Tables 2 and 4 were added to the Results section to illustrate the differences.

  1. There are several other minor comments that could be found in the attached file

Sorry, no file has been attached. 

Reviewer 3 Report

The paper addresses an issue of high interest: the impact of microplastics on aquatic organisms, particularly fish larvae. The draft is well written, the objectives are clear, the design is appropriate and the investigation methods are detailed correspondingly. 

The results are clearly presented and the discussions are oriented towards comparison with correspondent papers from the scientific literature. 

There are no significant drawbacks regarding the content. Minor text errors are present (ex.  closing ”)” in line 160).

Author Response

Dear Reviewer,

Thank you very much for appreciating our work.

The error in line 160 was corrected and the entire text was carefully revised.

Round 2

Reviewer 1 Report

My concers have been solved.

This revision is much better than the original one.

The English language is adequate.